# Analysis of Fast Fluorescence Kinetics of a Single Cyanobacterium Trapped in an Optical Microcavity

**DOI:** 10.3390/plants12030607

**Published:** 2023-01-30

**Authors:** Tim Rammler, Frank Wackenhut, Johanna Rapp, Sven zur Oven-Krockhaus, Karl Forchhammer, Alfred J. Meixner, Klaus Harter

**Affiliations:** 1Institute for Physical and Theoretical Chemistry, University of Tübingen, 72076 Tübingen, Germany; 2Center for Plant Molecular Biology (ZMBP), University of Tübingen, 72076 Tübingen, Germany; 3Interfaculty Institute of Microbiology and Infection Medicine, University of Tübingen, 72076 Tübingen, Germany

**Keywords:** cyanobacteria, photosystem, fast fluorescence kinetics, optical microcavity, fluorescence microscopy

## Abstract

Photosynthesis is one the most important biological processes on earth, producing life-giving oxygen, and is the basis for a large variety of plant products. Measurable properties of photosynthesis provide information about its biophysical state, and in turn, the physiological conditions of a photoautotrophic organism. For instance, the chlorophyll fluorescence intensity of an intact photosystem is not constant as in the case of a single fluorescent dye in solution but shows temporal changes related to the quantum yield of the photosystem. Commercial photosystem analyzers already use the fluorescence kinetics characteristics of photosystems to infer the viability of organisms under investigation. Here, we provide a novel approach based on an optical Fabry–Pérot microcavity that enables the readout of photosynthetic properties and activity for an individual cyanobacterium. This approach offers a completely new dimension of information, which would normally be lost due to averaging in ensemble measurements obtained from a large population of bacteria.

## 1. Introduction

Photosynthesis by cyanobacteria is one of the most important processes on earth. It was responsible for the oxygenation of the Earth about 2.4 billion years ago, which enabled the subsequent development of multicellular life forms [1]. Even today, cyanobacterial photosynthesis significantly contributes to oxygen (O_2_) generation during the process of carbon fixation typically from CO_2_ [2]. Furthermore, ancient cyanobacteria were the precursors of endosymbiotic chloroplasts [3] that enable all photosynthetic eukaryotes (including higher plants) to produce O_2_ and the chemical energy equivalents needed to fix CO_2_. Due to the importance of this process, and with the hope of replicating this process in organic solar cells, it is compelling to understand this process in its entirety. Although there is much already known, there are still unexplained phenomena in the photosynthetic process. One example is the energy transfer in photosystem 2 (PS2) from the absorption of a photon in the peripheral pigments to the reaction center, where the energy is used to split water. It is still not conclusively understood why the efficiency of this process is as high as 99% and whether this is related to non-trivial quantum optical effects [4].

The study of photosynthetic organisms by chlorophyll fluorescence kinetics has become an effective method for detecting even small changes in the photosynthetic process [5]. Among other techniques, this non-invasive method is already routinely used for the analysis of photosynthetic microorganisms, plant cell cultures and whole plants [6]. In this work, we present a novel method to study the photosynthetic efficiency in a living individual cyanobacterium on the basis of the well-studied Kautsky effect. In it, the kinetic behavior of the fluorescence emission is used to determine the quantum yield [7]. In contrast to previous analyses performed at the population level, our method enables one to determine the photosynthetic activity of an individual cell, exposing effects that are usually lost due to population averaging. 

To enable the very detailed analysis of the Kautsky effect, and thus, the photosynthetic efficiency at the single cell level, living cyanobacteria were placed in an optical Fabry–Pérot microcavity, as in our earlier work [8]. There, we focused more on the design of the experimental system, to which we would refer the reader for more technical details on the microcavity. In short, such a microresonator is composed of two semi-transparent mirrors, with a mirror separation of ~2 μm giving rise to microcavity resonances in the visible spectral range. In addition to strong coupling, a microcavity can also increase or decrease the spontaneous emission rate of a chromophore by tuning the resonance of the microcavity to the fluorescence of the chromophore or away from the fluorescence. This effect is called the Purcell effect [9,10,11] and shortens the fluorescence lifetime (FLT) of the chromophore for an on-resonant microcavity and prolongs it for an off-resonant microcavity [8]. Due to the small space inside the microcavity, it is difficult or impossible to use near-probe dependent means, such as O_2_ detectors, to determine the quantum yield in vivo. In order to capture the true photosynthetic efficiency of PS2, the experimental setup was designed as to minimize and eliminate residual phycobilisome (PBS) and photosystem 1 (PS1) fluorescence signals [12]. Our experimental approach offers a completely new way of obtaining knowledge about the PS2 activity, which would normally be lost due to averaging in ensemble measurements using a large cyanobacterial population.

## 2. Results

### 2.1. Theoretical Introduction and Technical Challenges

In our experiments, we investigated the unicellular cyanobacterium *Synechocccus elongatus* strain PCC 7942 (*S. elongatus*), a widely used model organism for photosynthetic research [13,14]. We have determined the quantum yields of individual cyanobacteria by analyzing the fast fluorescence kinetics (FFK) of PS2. The principle is based on the Kautsky effect [7], which describes the variable fluorescence (F_V_) after dark adaptation as a function of the irradiation time with actinic light (photosynthesis activating light) (see Figure 1A). The Kautsky effect divides the temporal evolution of the fluorescence signal into two regions: a short, initial time window of the fast fluorescence increase in intensity (fast fluorescence kinetics, FFK) occurring in microseconds (see Figure 1A,C,D, blue area), and the subsequent region of slow fluorescence kinetics (SFK), which describes the slow decrease occurring in seconds to minutes (see Figure 1A,C,D, green area). The photosystem of *S. elongatus* was spectrally analyzed to ensure that, according to the Kautsky effect, only the PS2 core antenna chlorophyll fluorescence was measured. The absorption spectrum of *S. elongatus* cells has four distinct bands: the chlorophyll a soret band at 440 nm [15], the carotenoid band at 500 nm [16], the PBS band at 630 nm [17], and the Q_y_ band of chlorophyll a at 680 nm [18] (Figure 1B). 

Excitation with 440 nm laser light as an actinic source (exciting the soret band) was most efficient and guaranteed almost exclusive fluorescence of chlorophyll a (compare to Figure 1B). Note that the laser source was operated in continuous wave mode, as pulsed laser sources might introduce undesired effects/damage. Conveniently, the soret band is spectrally distant from the chlorophyll a emission, which simplifies the separation of the laser scattering at 440 nm and the emission signal at 680 nm. In addition, a bandpass filter (676/29 nm) was placed in front of the detector to ensure that almost only photons emitted by PS2 were detected, since PS1 fluorescence occurs above 700 nm [19] and PBS fluorescence under 660 nm. Under these conditions, we can assume that only the fluorescence emission of the core antenna complex of PS2 was detected [20]. As the only type of chlorophyll in *S. elongatus* is chlorophyll a [21], the detected signal is dominated by chlorophyll a from PS2. 

In the following, we briefly explain the temporal evolution of a typical fluorescence signal (F_V_) and introduce three important parameters: minimum fluorescence (F_0_), maximum fluorescence intensity (F_M_), and terminal fluorescence (F_T_) [22]. Almost instantly, within a few picoseconds after actinic light irradiation, the photosynthetic chlorophyll fluorescence reaches an initial level called minimum fluorescence, F_0_ (Figure 1D). Since the transfer of the photon energy via dipole–dipole interaction is much faster than the electron transfer by a sequence of redox reactions, a kind of energy pile-up occurs in the physical to the chemical energy transformation. The absorbed energy initially leads to a fast increase in the fluorescence intensity to the level F_M_. F_0_ is thereby the maximum energy that can be emitted without an electron backlog. If the photosystem is irradiated with an intensity too low to induce closure of PS2 centers, F_0_ remains constant after the abrupt increase to the F_0_-level [20]. To reliably determine the maximal F_0_ value, all PS2 reaction centers must be open and the corresponding electron acceptor (i.e., first stable acceptor, plastoquinone, Q_A_) oxidized. This state can be achieved by 15 min of dark adaptation of the cyanobacterial cells [23]. After the abrupt rise to F_0_, the variable fluorescence (F_V_) rises to the maximum fluorescence F_M_, since there is a temporal delay between the first absorption of a photon after dark adaptation and the start of the carbon reduction cycle. During the rise from F_0_ to F_M_, the electron acceptor Q_A_ becomes increasingly reduced [24] and an electron backlog occurs because reduced quinone (QH_2_) cannot be re-oxidized fast enough and excess energy is released via fluorescence. Therefore, F_V_ reflects the redox state of Q_A_. F_M_ is only achieved when the actinic light intensity completely saturates all photosystems. In this case, all PS2 reaction centers are assumed to be in a closed state, and all associated Q_A_ are completely reduced. If the actinic light is not saturating, the peak intensity does not reflect the maximum possible fluorescence of the system. Here, it is important to ensure that F_M_ is reached after a few milliseconds. To prevent the delay of F_M_, the experiments were carried out in the presence of 3-(3,4-dichlorophenyl)-1,1-dimethylurea (DCMU). DCMU blocks the plastoquinone binding site (Q_B_) and thereby prevents electron transport from Q_A_ to Q_B_ [25]. Consequently, all Q_A_ sites become completely reduced after the onset of actinic light (Appendix A). This procedure enables one to determine the true F_M_ value, since the energy absorbed by PS2 is completely emitted radiatively due to blocking of further electron transport. In the absence of DCMU, F_V_ decreases again as soon as Q_A_ becomes re-oxidized due to electron transport towards NADPH/H^+^, essentially driven by PS1 activity. After a few minutes, it reaches the steady-state level F_T_, as illustrated in Figure 1A. This decrease is not possible in the presence of DCMU, and F_M_ remains constant. In this case, F_M_ and F_T_ are technically the same (Figure 1C).

The analysis of the FFK enables us to obtain precise information about the intactness and efficiency of PS2 [26]. A good approximation for the current photochemical efficiency can be calculated by using F_0_ and F_M_:


(1)
$$\Phi_{\mathrm{maxPS}2}=\frac{F_{M}-F_{0}}{F_{M}}=\frac{F_{V}}{F_{M}}$$


The current photochemical efficiency or quantum yield of PS2 Φ_maxPS2_ is a measure of the efficiency with which the excitation energy from the internal antenna pigments is used for photochemical reactions in the open P680 reaction centers [26]. Experiments that have determined the amount of oxygen produced as a function of light intensity show that Φ_maxPS2_ reflects, to a good approximation, the maximum relative electron transport efficiency [27]. In most plant species, Φ_maxPS2_ of about 0.83 can be expected [28]. However, determining Φ_maxPS2_ in cyanobacteria is not as straightforward as in plants, since additional antenna proteins, such as PBS, interfere with the measurement, distort the result and are the main reason for lower Φ_maxPS2_ yields [29]. After careful elimination of all interfering factors described above (DCMU application, narrow-band 440 nm laser excitation, use of a bandpass filter), a value for Φ_maxPS2_ of around 0.8 can also be expected for *S. elongatus* [29,30]. 

The major challenge of the current work is to detect the fluorescence signal of a single cyanobacterium which is embedded in an optical microcavity. For this reason, the microcavity with the embedded cyanobacterium was mounted on the scanning stage of a confocal microscope, and one bacterium was centered in the focal volume of the high numerical aperture (NA) objective lens. The large NA ensures that a large fraction of the photons emitted by the excited bacterium can be collected. At a laser power of 1.28 nW, a single bacterial cell emitted statistically only 60 photons per millisecond at the fluorescence maximum (F_M_), measured after the objective lens was put in place but before the cover slip was, on which the cyanobacterium was placed. Only about half of the bacterium was illuminated by the diffraction limited excitation spot. In addition, to resolve the sudden fluorescence increase (F_0_) in the time domain, highly sensitive and fast-responding detectors must be used (rise time < 2 ns; for more details, see Material and Methods). Due to the low photon emission rate (compare Figure 1D), a computer-assisted evaluation was necessary to carry out the data analysis. To determine the fast increase in the F_0_ emission, the raw data set was therefore smoothed by a Gaussian filter (Figure 1D, blue line). The maxima of the first and second derivatives of the smoothed data set indicate F_0_, since it is located in between these two extrema. The fluorescence maximum, F_M_, was obtained by averaging all values in a temporal window ranging from 0.08 to 1.00 s after F_0_ (Figure 1D, red dashed line indicates the start of a temporal window).

### 2.2. Measurement Results

The quantum yield of individual cyanobacteria was measured and analyzed via their individual fluorescence responses. For this purpose, the 440 nm laser (continuous wave mode) was precisely focused on one bacterium. The cross-section of a bacterium was approximated by a 1 × 2 μm ellipse. As a guide for the optimal light intensity for the following experiments, we used the standard setting for commercial chlorophyll fluorescence curve analysis devices of approx. 3000 μmol photons · s^−1^ · m^−2^ [28], which is equivalent to a 440 nm laser power of 1.28 nW per bacterium. To test for any effect in the variations of irradiation intensity on the quantum yield, FFK curves of different individual bacteria from the same growth culture were measured with different laser intensities (see Figure 2B,C; 3 and 90 nW).

As shown in Figure 2B, the quantum yields of individual bacteria were the same for laser exposure of 90 and 3 nW, as the difference in their medians lies within the two statistical distributions (two-tailed *t*-test, *p* = 0.343). This indicates that the difference in the quantum yield cannot be attributed to unequal irradiation conditions. Under these conditions, a change in the quantum yield could therefore only be caused by the nature of the surrounding electromagnetic field. In order to test the lower limitation of the FFK in a single cell, the quantum yield of chlorotic cyanobacteria was recorded. The chlorosis of the cyanobacteria was induced by nitrogen starvation, which induces the degradation of photosynthetic pigments, in particular, those of the PCBs, whilst the bulk of the photosystems stays intact [31]. As shown in Figure 2D,E, the quantum yield (excitation 440 nm, 1.28 nW) in chlorotic cyanobacteria was significantly lower ($\Phi_{\mathrm{maxPS}2}=0.74$), compared to the yield from non-chlorotic cells ($\Phi_{\mathrm{maxPS}2}=0.77$). 

An advantage of the single-bacterium experiment over the classical population approach is that also the distribution of the quantum yields can be analyzed statistically and individually. The possibility of determining the quantum yield of a single bacterium allows one to investigate extreme examples (outliers) in more detail. In a population experiment, small-scale differences of individual cells would be averaged. 

An example in which a single-bacterium approach is advantageous for FFK analysis is a situation where the photosynthetic light conditions vary on a small spatial scale, which is the case in an electromagnetic Fabry–Pérot microcavity. Here, each cyanobacterium experiences a different optical field. Fabry–Pérot microcavities consist of two semitransparent mirrors (quality factor, Q = 98) with an optical path length allowing resonances in the visible spectral range. Since the upper mirror is curved, the mirror spacing varies depending on the spatial location, which is why the individual cyanobacteria experience different electromagnetic fields. For analysis in the microcavity, the cyanobacteria were embedded in a BG-11 agarose matrix to restrict spatial drift [8]. In the set-up, the cyanobacteria located in the microcavity were irradiated with 440 nm laser light from below and the residual fluorescence recorded through a high-NA objective lens (NA = 1.4) (see Figure 3A). More details about the experimental setup were published before [8] and are also given in Materials and Methods.

In the first series, the FFK measurements were performed in free space (see Figure 3C,D, green). The second series was recorded in a microcavity resonant to the absorption/emission at 680 nm (see Figure 3C,D, red; and illustration in Figure 3B, right) and in the third series in an off-resonant microcavity (see Figure 3C/D, blue and illustration in Figure 3B, left) with resonance set to approx. 500 nm. The excitation energy was 1.28 nW laser intensity). The largest quantum yield was observed in free space and in the off-resonant microcavity. In contrast, there was a significant difference between the quantum yield in the off-resonance and the on-resonance (*p* = 1.52 × 10^−4^) microcavity, and between free space and the resonant (*p* = 7.54 × 10^−8^) microcavity (Figure 3C). Detailed values of the absolute, F_0_-normalized, and F_M_-normalized data of quantum yield measurements in free space, resonant, and off-resonant microcavity are given in the Appendix A. Hence, only the resonant microcavity had a significant reductive impact on the quantum yield of single cyanobacteria, which also corresponds to previous fluorescence lifetime measurements of microcavity-enclosed cyanobacteria [8]. Since the same ambient conditions prevail in the free space and resonant microcavity, we can assume that the quantum yield is solely influenced by the special light conditions, i.e., the coherence of the optical field inside the resonant microcavity.

## 3. Discussion

We have developed a method to reliably determine the quantum yield of individual living cyanobacteria and demonstrated the reliability of the method. Moreover, we showed that the measurements tolerate slight variations in the excitation intensity of the actinic light. To our opinion, the approach applied here and in our previous publication [8] are the first examples showing that an optical cavity enables the interrogation of biological systems in terms of quantum physical phenomena, as theoretically proposed recently [32].

An especially interesting scenario is observed when the microcavity resonance coincides with the fluorescence and absorption maxima of the cyanobacterium (see Figure 1B), where the excitation energy can coherently oscillate back and forth between the electromagnetic field in the microcavity and the photosynthetic pigments. This leads to so called strong light–matter coupling, resulting in a polaritonic state, which manifests itself as a double-peaked microcavity transmission spectrum with a peak spacing called vacuum Rabi splitting [33].

The spread in the single-cell experimental data most likely originated from the biological variation in the photosynthetic quantum yields of individual whole cyanobacteria. Impressively, our method can be used to show such differences. The magnitude of the spread might also allow deriving inferences about other biological effects, such as non-obvious deficiency symptoms or suboptimal environmental conditions of a bacterial culture. Our highly spatially resolved and direct measurements also demonstrate that cyanobacteria differ in their photosynthetic activity in the optical microcavity. Since the quantum yields differ significantly between a resonant and an off-resonant microcavity, it is reasonable to propose that the coherent optical field in the microcavity has a significant effect on the energy transfer in PS2 in vivo. 

Our method allows us to study the microcavity’s influences on a single cyanobacterium. We are of the opinion that such a method will be helpful for future researchers to unravel remaining open questions of photosynthesis, e.g., about the efficiency of optical-to-chemical energy conversion.

## 4. Materials and Methods

### 4.1. Experimental Set-Up

A home-built confocal scanning microscope was used for the measurements in this work (compare Figure 4). The excitation light source is a laser diode (LDH-D-C-440, PicoQuant GmbH, Berlin, Germany) operated in continuous wave mode with an excitation wavelength of 440 nm. The beam is directed via two mirrors through a clean-up interference filter (MaxDiode™ LD01-439/8-12.5, Semrock Optical Filters, Rochester, NY, USA) onto a lens, which combines the beams into a single-mode glass fiber (P1-405 BPM-FC-2, Thorlabs Inc., Newton, NJ, USA). After the decoupling unit, the excitation light is passed through various gray filters and then onto a swiveling lens. This lens can be used to focus on the back aperture of the objective lens to acquire widefield images. After the lens, the beam is reflected by a dichroic beam splitter, which is positioned at a 45° angle to the propagation direction (F48-487 Laser Beam Splitter zt 488, AHF Analysentechnik AG, Tübingen, Germany). The beam splitter reflects light with wavelengths below 488 nm, whereas it transmits light with longer wavelengths. The reflected beam is now focused on the sample through an oil objective lens (Zeiss Plan-Apochromat, 100×, 1.4 Oil DIC, Carl Zeiss Jena GmbH, Jena, Germany). Micrometer screws are used for coarse positioning of the sample, and a piezo-controlled scanning stage (P-733.3CD, Physik Instrumente (PI) GmbH & Co. KG, Karlsruhe, Germany) allows sample scanning in three spatial dimensions. The fluorescent light emitted by the sample is collected by the same objective lens and transmitted through a beam splitter. An additional long-pass filter (488 LP Edge Basic long-pass filter, BLP01-488R-25, Semrock Inc., USA) and a band-pass filter (BrightLine^®^ FF01-676/29-25, Semrock Inc., USA) are used to filter the detected signal. Confocal images of the fluorescence intensity are acquired by a single photon avalanche diode (SPAD) (PDM Series, Micro Photon Devices, Bolzano, Italy). The SPAD is coupled with a TCSPC unit (HydraHarp 400, PicoQuant GmbH, Berlin, Germany) for the acquisition of time-correlated fluorescence traces. Spectra are acquired with a spectrometer (Acton SP300i, Princeton Instruments, Trenton, NJ, USA) with a thermoelectrically cooled CCD camera (PIXIS 100, Princeton Instruments, Trenton, NJ, USA). With widefield illumination, videos or images can be recorded in real time. The scanning stage, the SPAD, the laser diode and the TCSPC unit are controlled by SymphoTime^®^ software 64 (PicoQuant GmbH, Berlin, Germany). Fluorescence spectra are acquired with the software Winspec^®^ (Princeton Instruments, Trenton, NJ, USA). This software controls the monochromator and the corresponding CCD camera. 

### 4.2. Preparation of Microcavity Mirrors

The mirror preparation was achieved by evaporating a 3 nm chromium layer on a glass cover slip to ensure that the following layers adhere well. Next, the reflective silver layer with a thickness of 30 or 60 nm—for the lower and upper mirrors, respectively—was vapor-deposited. A gold layer (5 nm) and an SiO_2_ layer (20 nm) served as a coating layer, since silver is very susceptible to oxidation and damage and is bactericidal [34]. The structure of these layers created a microcavity with a quality factor of Q = 98. The distance between the two mirrors was adjusted very precisely by a piezo actuator. The custom-built holder was mounted on the stage of a scanning confocal microscope to measure the fluorescence of cyanobacteria one at a time.

### 4.3. Light Intensity Measurements

The light intensity for the cultivation for *S. elongatus* was measured with a Li-Cor Li-189 radiometer from Heinz Walz GmbH (Pfullingen, Germany). The laser power was measured with an Optical Power Meter Model 1830-C and a Sensor Model 818-SL, both from Newport Corporation (Irvine, CA, USA). 

### 4.4. Bacterial Cultivation Conditions

*Synechococcus elongatus* PCC 7492 was cultivated under photoautotrophic conditions under continuous light with an intensity of around 30 μmol photons · s^−1^ · m^−2^ (Lumilux de Lux, Daylight, Osram, Munich, Germany). The cultivation was performed in 50 mL BG11 medium [35] supplemented with 5 mM NaHCO_3_ in 100 mL Erlenmeyer flasks at 28 °C and continuous shaking (120–130 rpm). 

Nitrogen-starved, chlorotic cells were obtained as previously described [36] with slight modifications. Exponentially grown *S. elongatus* cells in BG11 were harvested by centrifugation at room temperature (3500× *g*, 10 min), the supernatant was discarded, and the cell pellet was washed twice with 50 mL NaNO_3_-free BG11-medium (BG11-0). After that, the cultures were cultivated in BG11-0 at light intensities of 50–60 μmol photons · s^−1^ · m^−2^ for 1 day.

### 4.5. Bacteria Preparation

The cyanobacteria were treated with a 10 μmol 3-(3,4-dichlorophenyl)-1,1-dimethylurea (DCMU) solution. Then, 20 μL of a *S. elongatus* suspension was embedded in a low-melting agarose matrix (prepared with BG11 [35] medium) to prevent cell movement.

## Figures and Tables

**Figure 1 plants-12-00607-f001:**
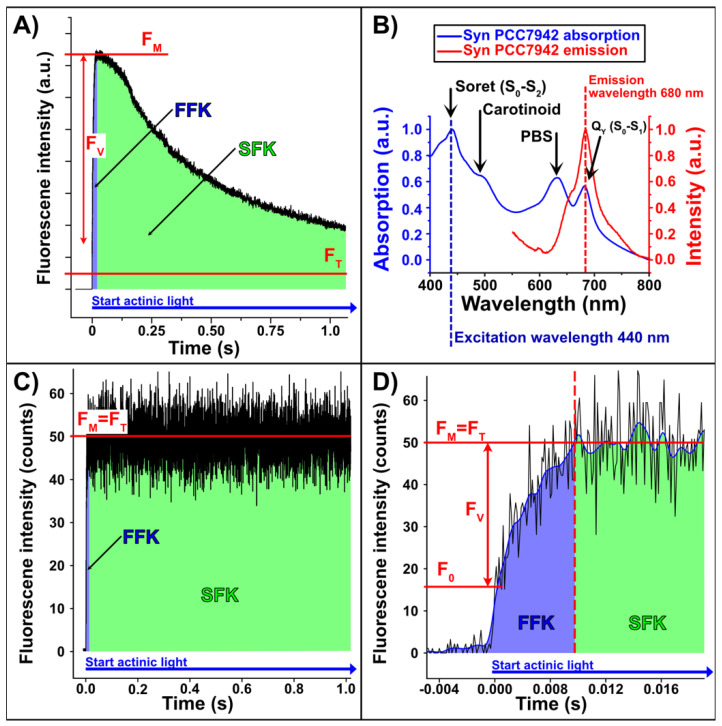
(**A**) Illustrative example of a fluorescence intensity time trace of a *S. elongatus* population (without any special treatment) excited with saturating light intensity (blue arrow). Shown in red: the maximum fluorescence intensity (F_M_) and the terminal fluorescence (F_T_) intensity according to the Kautsky effect. The blue area indicates the temporal range of the fast fluorescence kinetics (FFK), and the green area indicates the range of the slow fluorescence kinetics (SFK). (**B**) Normalized fluorescence (red) and absorption (blue) spectra of *S. elongatus* measured in BG11 medium, taken from [8]. The bacteria were excited with 440 nm light (blue dashed line). This spectrum was acquired with a conventional UV/VIS spectrometer. In contrast to the following measurements, no spectral filters were used for wavelength selection. The bacteria have both absorption and emission maxima at approx. 680 nm (red dashed line). (**C**) Fluorescence intensity time trace of *S. elongatus* after treatment with 3-(3,4-dichlorophenyl)-1,1-dimethylurea (DCMU). By spiking with DCMU, F_T_ no longer decreases to the same level as shown in (**A**), because electron transport between PS2 and PS1 is blocked, and the excitation energy is released almost exclusively by fluorescence. In this case, the terminal fluorescence (F_T_) is equal to the maximum fluorescence (F_M_). (**D**) Expanded section of (**C**) to determine the sudden fluorescence increase F_0_ and the maximum fluorescence F_M_. These values are used to quantify the quantum yield (Φ_maxPS2_ = 0.7 for this particular example).

**Figure 2 plants-12-00607-f002:**
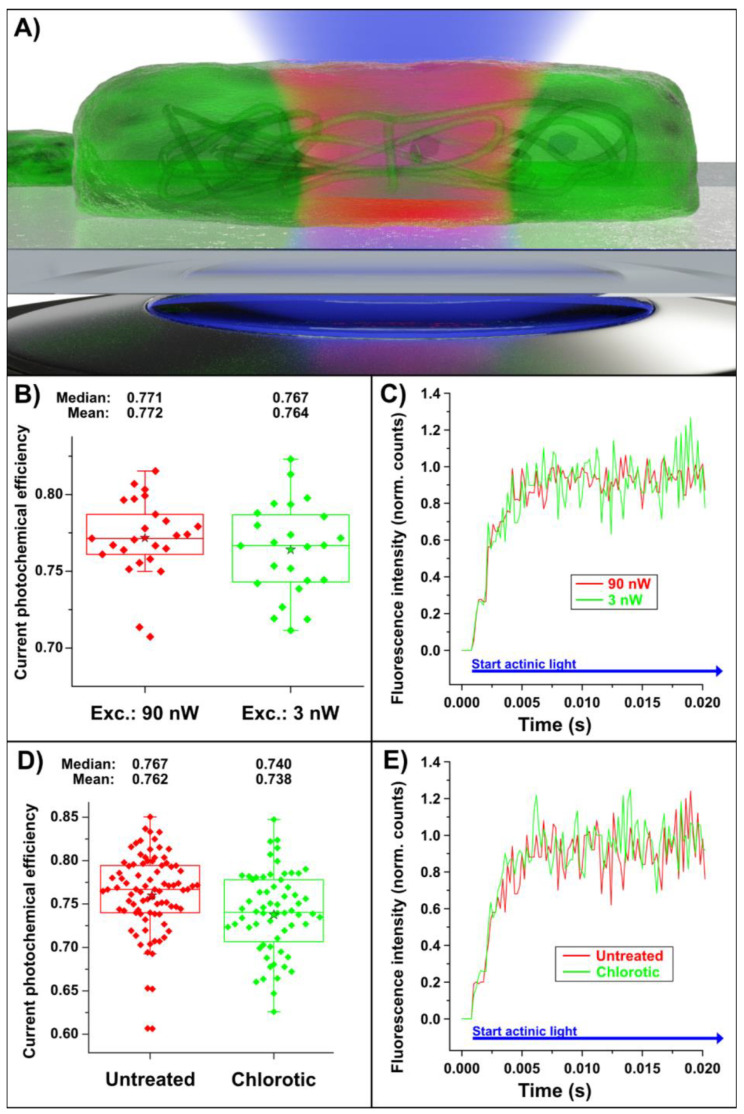
(**A**) Schematic illustration of a single bacterium on a glass coverslip in the focal volume of a high-numerical-aperture (NA = 1.4) objective lens. (**B**) Box plot of the quantum yield of *S. elongatus* irradiated by 90 nW (red dots) and 3 nW (green dots) laser power to prove the independence of the photosynthetic efficiency from the laser power. There is no significant difference between 90 and 3 nW (two-tailed *t*-test, *p* = 0.343). (**C**) Two representative fluorescence curves recorded with a laser excitation intensity of 90 nW (red curve) or 3 nW (green curve). The curves were normalized to F_M_ because different amounts of chlorophyll were measured due to different bacterial sizes, allowing a direct comparison. (**D**) Box plot of the effective quantum yield of *S. elongatus* in free space (red curve) and chlorotic cells (nitrogen starved) in free space (green curve). Statistically, the median of the quantum yields of the free space and the chlorotic bacteria differ significantly (two-tailed *t*-test *p* = 3.10 × 10^−3^). (**E**) Two representative fluorescence curves (normalized to F_M_) of untreated cyanobacteria (red curve) and cyanobacteria grown under chlorotic conditions (green curve). The star in (**B**) and (**D**) represents the mean.

**Figure 3 plants-12-00607-f003:**
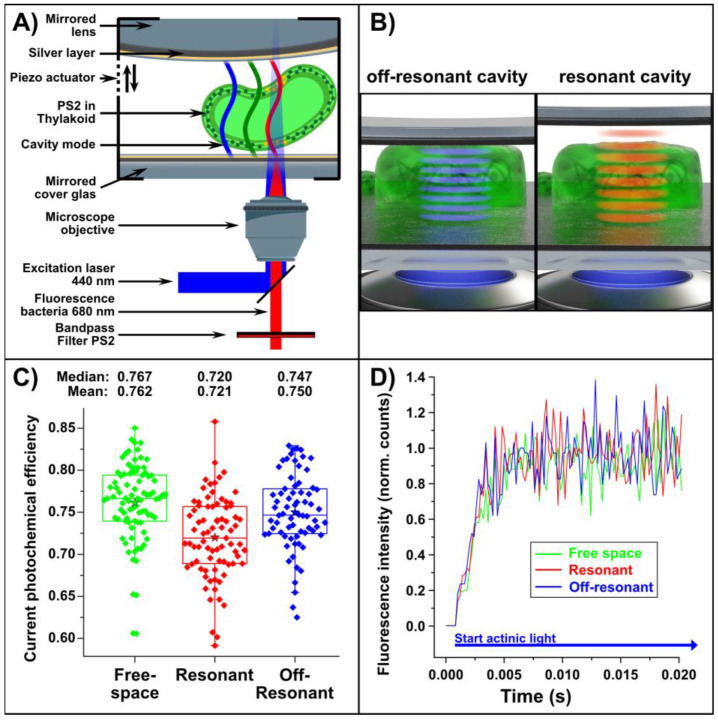
(**A**) Layout of the experimental set-up. A laser (440 nm, continuous wave) is focused by an objective lens onto a single bacterium in the microcavity that consists of two semi-transparent silver mirrors. The fluorescent light of the bacterium is transmitted by the beam splitter and directed to a detector. (**B**) Illustration of *S. elongatus* in a microcavity (consisting of two very close silver mirrors). On the right, bacterial fluorescence is resonant with the microcavity mode (red waves), which leads to strong coupling and splitting of the energy levels. For detailed measurements, please refer to: [8]. On the left, *S. elongatus* is illustrated in an off-resonant microcavity. (**C**) Box plot of the quantum yield of *S. elongatus* in free space (green dots), in a resonant (red dots) and an off-resonant (blue dots) microcavity. The quantum yield drops significantly inside the resonant microcavity (two-tailed *t*-test: compared to free space: *p* = 7.54 × 10^−8^; compared to off-resonant microcavity: *p* = 1.52 × 10^−4^), with otherwise equal ambient conditions in both series of measurements. Statistically, the free space and off-resonant measurement series do not differ (two-tailed *t*-test, *p* = 0.12). (**D**) Three representative fluorescence curves (normalized to F_M_) of cyanobacteria in free space (green curve), in a resonant microcavity (red curve) and an off-resonant cavity (blue curve). The star in (**C**) represents the mean.

**Figure 4 plants-12-00607-f004:**
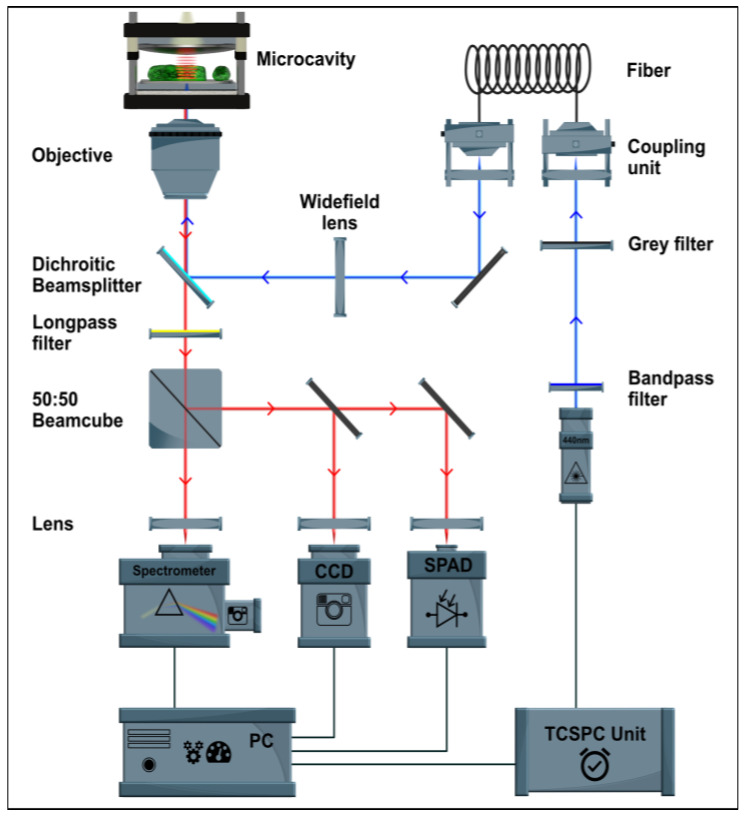
Home-built semi confocal microscope.

## Data Availability

Data are available in the main text or the Appendix A. Further material is available from the corresponding author upon request.

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
