# Peer review of "Analysis of Fast Fluorescence Kinetics of a Single Cyanobacterium Trapped in an Optical Microcavity"

_plants, 2023, doi:10.3390/plants12030607_

Round 1
Reviewer 1 Report
Rammler and co-authors present a novel fluorescence-based method to measure the efficiency of photosystem 2 (PSII) in individual cyanobacteria cells. The authors have optimised the method to measure accurately the minimal and maximal fluorescence of PSII while minimising fluorescence contributions from phycobilisomes (PBS) and photosystem 1 (PSI). This allows measuring the maximal PSII quantum yield (ΦmaxPS2), that is notoriously difficult to quantify correctly in vivo in cyanobacteria. By allowing the quantification of ΦmaxPS2 in individual cells, the method provides information on the biological variability between them. This information is normally lost when measuring photosynthetic parameters in bulk cell cultures. The experimental design is appropriate and the results are scientifically sound and correctly presented.
As the authors clearly state in the Discussion, the method can give new insights on the biological variability of the photosynthetic efficiency in function of environmental conditions. Additionally, the use of a resonant microcavity can be useful to investigate the optical effects potentially affecting the efficiency of the use of excitation energy for photochemistry. The novelty of the method and the opportunities it offers to gain a better understanding of the photosynthetic efficiency of cyanobacteria are going to be of interest for the scientific community. For these reasons, the work presented here deserves publication. Nonetheless, some parts of the text require substantial revision to rectify incorrect or misleading statements and improve clarity. There are also problems with the references and supplementary information.
Three major points require attention:
1- Description of fluorescence parameters
L118-121: “If the photosystem is irradiated with an intensity too low to induce photochemistry, F0 remains constant after the abrupt increase to the F0-level, hence F0 is independent of photochemical processes”. This is incorrect, because F0 strictly depends on the fact that PSII centres are in an open state and, therefore, are able to quench the excited state of chlorophyll by doing photochemistry, outcompeting the radiative decay of the excited state. If there were no photochemistry, the system would be at Fm. F0 remains constant if the light is low enough not to induce the “electron backlog”, i.e. the excitation rate is slower than the rate of QA- reoxidation. Here the authors refer to a book chapter written in German, so it is difficult for non-German speakers to retrieve information on where the statement comes from. An alternative review explaining the fundamentals of fluorescence is “Chlorophyll Fluorescence and Photosynthesis: The Basics. G H Krause, and, and E Weis, Annual Review of Plant Physiology and Plant Molecular Biology 1991 42:1, 313-349” (currently Ref. 21, although incomplete). This is a major theoretical error that needs to be corrected.
L122-127: the authors state that to measure F0, cyanobacteria must be dark-adapted for 15 minutes, because shorter or longer dark times distort F0 due to quenching effects. It is unclear how this time has been determined, and which quenching effects the authors refer to. Again, here they refer to Ref. 24, which is in German and does not seem to be about cyanobacteria. The authors need to either show experimental evidence of how the level of F0 changes in function of the dark adaptation time or provide a correct reference and elaborate on the mentioned quenching effects (taking into account that these can vary with the cell’s physiology).
In L137-143 the authors state that DCMU is used to measure Fm, to ensure the full reduction of QA within few milliseconds from the actinic light onset, which is perfectly fine and reasonable. Still, Fm can be reached also in absence of DCMU with a saturating illumination, although it is true that the use of DCMU removes all doubts about it being reached. The authors themselves show this in Figure S1, reporting similar ΦmaxPS2 in presence and absence of DCMU. This figure should be mentioned here. Then in L143-145 the authors say that the reaching of Fm can be delayed by protective dissipation mechanisms when cyanobacteria are illuminated with high light intensities, but it is not clear how this applies here, and whether this is supposed to happen in presence of DCMU or not. Dissipation mechanisms need time to be activated, so they should not influence the reaching of Fm within the first few milliseconds of illumination. Here the authors refer to Ref. 27, which is about CAM plants, so clearly not correct. This part is confusing, and should be either explained, clarifying how this applies here, or removed.
2- Description of the experimental conditions
L100-102: the authors state that excitation with a 440 nm laser light induces almost exclusively fluorescence of PSII, but this is not supported by the emission spectrum shown in Fig. 1B. By exciting chlorophyll a, the 440 nm light also excite PSI, whose fluorescence can be seen a broadening of the red part of emission spectrum (cf. Fig. 1B with the Fv spectrum of S. elongatus 7942 in Fig. 5L of Ref. 31). How do the authors conclude that fluorescence from PSI is only 2%? Additionally, the emission spectrum in Fig. 1B clearly contains contribution from PBS, seen as a fluorescence shoulder at 650-660 nm. It is true that these contributions are eliminated by the concomitant use of the 440 nm excitation and the 676/29 nm bandpass filter, but this section should be re-written to make it consistent with the experimental evidence.
L203-2011: the authors first say that they use an excitation of approx. 3000 µmol photons s-1 m-2, corresponding to a 440 nm laser power of 1.28 nm, but then in Fig. 2B they show the comparison between a laser power of 3 nW and 90 nW. At this point, it is not clear which intensity has been used for the other measurements (Fig. 1, 2D-E and 3C). Is it 1.28 nW or 3 nW? If 3 nW, how many µmol photons s-1 m-2 does it correspond to? This should be clarified. Please also note that in the legend of Fig. 2B the authors first write 90 and 3 nW and then 90 and 2 nW, which adds to the confusion. In L271 the authors say that the measurements tolerate “slight” variations in the actinic light intensity. Do they refer to the difference between 90 and 3 nW? How big is the difference in photon flux between the two intensities? Perhaps I am wrong, but this sounds like a big difference (which would not be a problem, but rather confirm the robustness of the results).
At some point of the experimental setup description, the authors should mention that a CW illumination has been used because of undesired damage induced by the use of a pulsed laser and refer to Fig. S2 and associated text, because this is not mentioned anywhere in the main text, but is very relevant especially with respect to the effects of a resonant microcavity.
3- References: they should be numbered in order of appearance, at the moment they are not (e.g. in Introduction there is 7, 9, 11, 12, 36, 8, and Ref. 10 only comes in the Discussion). The choice and appropriate use of references is also questionable. Examples:
L104: Ref. 17 is used regarding the large Stokes shift between excitation in the Soret and chlorophyll a emission, although the paper is about PBS degradation and does not contain fluorescence data.
L108-109: Why the Rippka article (Ref. 19) here and not when the cyanobacteria species is first mentioned?
Again, check the correctness of using Ref. 24 in L124-126 and Ref. 27 in L145.
These are only a few examples, the references should be checked thoroughly.
Other minor points:
- Fig. 1A, C and D: At a first glance, it is not very clear that the excitation light is switched on at the start and then kept on throughout the measurements. Perhaps the authors could include an arrow or something similar to mark the light onset, and mention it in the legend. Additionally, the fluorescence parameters marked in the figure should be described briefly in the legend (also stating after how much time FT is reached in panel A).
- Fig. 1B: the emission peak is indicated by a red dashed line, while in the legend this line is mentioned as being black. Both absorption and emission spectra are undistinguishable from those published by the same authors in Fig. 1B of Ref. 9. Perhaps the data are very reproducible, but if the spectra are the same, this should be stated.
- Fig. 4: “Objective” and not “objective”. Could arrowheads be included to show the direction of the light beams? It is not necessary, but it would improve the understanding of the setup.
- L32: the chloroplasts are not just in higher plants, but in all photosynthetic eukaryotes.
- L157: rather than “adopted”, the excitation energy is used for photochemistry by the open P680 reaction centres.
- L160-161: ΦmaxPS2 “of photosystem 2” is redundant.
- Fig. S2 has panel A but no panel B, and left and right are used in the legend.
Reviewer 2 Report
Tim Rammler et al. wrote the manuscript titled “Analysis of Fast Fluorescence Kinetics of a Single Cyanobacterium Trapped in an Optical Microcavity”. The authors have developed the new technique to observed chlorophyll fluorescence from a single cell by using an optical Fabry-Perot microcavity. While the optical system of the microscope was almost the same as the already reported one (ref. 9), in this time, the authors modified it to a confocal system and newly applied a cw laser instead of a pulse laser. The authors observed single cells of Synechococcus elongatus PCC 7942 cultivated under healthy or nitrogen-starved conditions and have tried to analyze the initial kinetics in an induction phase of chlorophyll fluorescence. The method and experiments are potentially original and interesting, but the authors seem to extract almost nothing for plant or photosynthetic science from the newly observed results. Therefore, the manuscript critically lacks interpretation and discussion of data and should be revised thoroughly before acceptance for publication in the journal “Plants”.

Round 2
Reviewer 2 Report
The revised version of manuscript is honestly improved to enable readers to easily outlook its scientific significance, whereas it remains one critical question on in the introductive data. In the figure 1, the FM level fell into about one third in the presence of DCMU. The authors stated that the true and maximum FM level is observed when QA are open in all the PS2 and should be achieved in the presence of DCMU. This statement is however apparently inconsistent with the one-third FM level in figure 1, which must be clarified before the acceptance.
